# Evaluation the validity and reliability of the perceived medical school stress scale in Turkish medical students

**Esra Çınar Tanrıverdi**[1]*, **Sinan Yılmaz**[2], **Yasemin Çayır**[3]

**1** Faculty of Medicine, Department of Medical Education, Atatürk University, Erzurum, Türkiye, **2** Faculty of Medicine, Department of Public Health, Atatürk University, Erzurum, Türkiye, **3** Faculty of Medicine, Department of Family Medicine, Atatürk University, Erzurum, Türkiye

* esracinart@yahoo.com

**Citation:** Çınar Tanrıverdi E, Yılmaz S, Çayır Y (2023) Evaluation the validity and reliability of the perceived medical school stress scale in Turkish medical students. PLoS ONE 18(8): e0288769. https://doi.org/10.1371/journal.pone.0288769

**Data Availability Statement:** All relevant data are within the paper and its Supporting information files.

**Funding:** The author(s) received no specific funding for this work.

## Abstract

Medical education can be a challenging and stressful process. Additional stressors can make the medical education process even more complex and impair a student's attention and concentration. To the authors' knowledge, there is no valid and reliable scale to measure medical school stress in Turkish medical students. Therefore, this study aimed to determine the validity and reliability of the Perceived Medical School Stress (PMSS) Scale in Turkish medical students. The Perceived Medical School Stress Scale is a self-assessment tool developed to measure medical school-induced stress in medical students. It consists of 13 items divided into two subdimensions. Scale items are answered using a four-point (0–4) Likert system The total score that can be obtained from the PMSS ranges from 0 to 52, with higher scores indicating higher levels of perceived stress. First, the scale was applied as a pilot to 52 students by performing the scale's back-and-forth translation into Turkish. Then, the scale was applied to 612 volunteer medical students to ensure validity. Convergent validity and confirmatory factor analysis are used to assess the construct validity of a scale. Test-retest, item correlations, and Cronbach's alpha coefficients are used to evaluate the reliability of a scale. As a result of confirmatory factor analysis, the two-factor structure of the original scale was confirmed. The fit indices of the model obtained showed excellent fit. The Generalized Anxiety Disorder-7 (GAD-7) Scale was used for convergent validity. The GAD-7 is a self-assessment tool that measures the level of generalized anxiety. It is answered with a four-point Likert scale for the last two weeks. The score that can be obtained from the scale is between 0–21. A score of ten or more indicates possible anxiety disorder. The students' mean perceived medical school stress score was 39.80±8.09, and their GAD-7 score was 11.0±5.5. A significant positive relationship was found between the total scores of the scales (r = .48, $P <$ .001). The Cronbach's alpha value of the scale was .81, and test-retest reliability was significant for all scale items ($P <$ .001 for all). No item was deleted according to Cronbach's alpha values and item-total correlations. There was no significant relationship between Turkish version of the PMSS and GAD-7 scores and age, sex, income status, tobacco use, or exercise ($P >$.05). The Turkish version of the Perceived Medical School Stress Scale is a valid and reliable scale that can be used to investigate the medical school-specific stress of students.

**Competing interests:** The authors have declared that no competing interests exist.

## Introduction

Medical education is a lifelong and demanding process that begins with medical school admission. The stress that medical education creates on students is a well-recognized issue that has been studied extensively in many countries worldwide [1–3]. Awareness of stress in medical education is also increasing in Türkiye [4,5].

Academic life can create a lack of time for other activities by affecting the personal and social life of individuals. Not being able to save enough time for their families, friends, social life, and personal development increases students' stress. University students can experience a range of stressors beyond the academic demands of their coursework, such as moving to another city for university, financial stress, and the need for shelter or a house [6].

Research has consistently shown that medical students experience higher stress, anxiety, depression, and burnout symptoms than students in other fields of study, peers, and the general population [1,3,7–9]. It is generally accepted that the high-stress levels experienced by medical students are due to a range of specific stressors unique to medical education [1,6,8]. Among these reasons are the intense curriculum and information overload, academic competition, high-performance expectations and grade anxiety [1], fear of inadequacy and making mistakes, and the faculty's insensitivity to students' needs [1,6,8,10,11]. In addition, attention is also drawn to emotional stressors, such as witnessing people's illnesses, disabilities, suffering, and deaths [1,8].

Intense stress is one of the leading causes of cognitive dysfunction [12]. The stress experienced by medical students negatively impacts their school performance, leading to a decrease in academic achievement [1], dropping out of school, and even suicidal thoughts [13,14]. The stress experienced by medical students is also closely related to burnout [12,15]. A recent study in Turkey showed that the psychological health of medical students deteriorated within the first year of medical education [16].

Perceived stress affects students' psychological health and attitudes toward patients. It has been reported that stress negatively affects medical students' ethical and professional attitudes and causes empathy loss toward patients [13,15]. Perceived stress thus negatively affects the quality of patient care [17].

Studies report that even though medical students are aware of the stress they experience, they are reluctant to accept their psychological problems, seek professional help, and even show resistance in this regard [18,19]. Therefore, to develop preventive strategies, it is necessary to recognize medical students' stress and identify the factors affecting it.

The Perceived Medical School Stress (PMSS) scale is an easy-to-answer and practical tool developed to measure medical school-specific stress perceived by medical students [6]. It has been adapted and widely used in various languages and cultures [1,8,10,20,21]. There is currently no version of the scale adapted to Turkish. In Türkiye, studies evaluating medical students' perceived stress levels use scales such as the "Perceived Stress Scale" and "Depression Anxiety and Stress Scale," which were validated in Turkish [16,22]. Although these scales assess stress, they cannot distinguish medical school-specific stress from general stress.

This study aims to adapt the PMSS scale to Turkish, conduct validity and reliability studies, and provide a unique tool to be used in evaluating medical school-induced stress of medical students in Türkiye.

## Materials and methods

### Study design and participants

This study is a methodological, two-stage observational validity study. The research was carried out with Atatürk University Faculty of Medicine preclinical students between July and

September 2022. Initially, Prof. Dr. Peter P. Vitaliano, who developed the original scale, was contacted via e-mail, and the necessary permission was obtained to adapt the scale into Turkish. Subsequently, the study was approved by the Atatürk University Faculty of Medicine Non-Invasive Studies Ethics Committee (Number: B.30.2.ATA.0.01.00/05, date: 24.06.2021). The study was carried out under the rules of the Helsinki Declaration. Informed consent from the participants was obtained.

Although 5–20 participants are recommended for each scale item in validity and reliability studies [23], all volunteering students' inclusion was aimed at the investigation since a more extensive study sample was preferred. At the time of the study, there were 1080 preclinical students in the medical faculty. Of these students, 612 volunteers participated in the study. In addition, 14 days after the first application, the Turkish version of the PMSS (PMSS-TR) was readministered to 70 randomly selected students who agreed to participate for the second time to evaluate the test-retest reliability.

Data were collected through an online survey. Students were first informed about the purpose and scope of the study via e-mail and classroom WhatsApp groups. Then, the survey link was shared with the students. Information about the purpose and scope of the study was also included at the beginning of the questionnaire. The first question of the questionnaire was written as "I have been informed about the study, and I accept participation voluntarily." Participants could not answer other questions without giving this consent. In this way, online permission was obtained from the participants. Data were collected anonymously, and students were not asked for personal information. However, the students were asked to write a nickname to match with the retest. The questionnaire was available to access for three weeks. During this period, a weekly reminder message was sent to the students. At the end of the period, the survey was terminated. Answering the questionnaire takes approximately 15–20 minutes. Students with a diagnosed psychiatric illness, those using a medication, and those who did not volunteer were excluded. No incentives were paid to the students for participation.

## Data collection tools

A three-part questionnaire was used as a data collection form in the study. The first part of the questionnaire contained questions about sociodemographic characteristics (age, sex, years of education, income status, tobacco use, and exercise), the second part had the PMSS scale, and the third part included the Generalized Anxiety Disorder-7 scale.

## Perceived Medical School Stress Scale (PMSS)

The PMSS was developed by Vitaliano et al. to measure the stress of medical students from studying in medical school. The original language of the scale is English. There are two dimensions and 13 items on the scale. There are nine items in the "Psychological Stress and Environment" dimension (items 1, 4, 6, 7, 8, 9, 10, 11, and 12) and three in the "Resilience and Expectations" dimension (items 2, 3, and 5). The 13th item was added to the scale by the authors' majority approval because it was a primary concern of students (finance). However, this item was not included in either of the two subdimensions. The scale consists of items such as "Medical education controls my life and leaves little time for other activities," "Medical school is cold, soulless, and unnecessarily bureaucratic," and "My financial situation is a concern for me," which questions various situations related to stress such as workload, financial concerns, competition, and threats [6].

The scale has a 5-point Likert scale. Scale items are answered and scored as "0 = strongly disagree" and 4 = strongly agree." The score obtained from each item is between 0–4. There is no reverse-scored item on the scale. The total score is obtained by summing the scores of each

item. The total score that can be obtained from the scale varies between 0–52. High scores are associated with higher perceived stress [6,8]. The Cronbach's alpha of the scale was .81 [6].

Scores between 1 and 5 were used in the scale's European versions. The total score obtained in these versions is between 13–65 [7,10,20].

### Generalized Anxiety Disorder Scale (GAD-7)

The GAD-7 is a tool developed by Spitzer et al. (2006) to measure anxiety levels. It inquires about symptoms in the past two weeks. Scale items are answered and scored according to a four-point Likert system (0 = almost never, . . .. 3 = almost every day). The score that can be obtained from the scale is between 0–21. The cutoff point of the scale was determined to be 10. A GAD-7 score of ≥10 indicates a high probability of generalized anxiety disorder. For total scores from the scale, ≥5–9, ≥10–14, and ≥15 are cutoff points for mild, moderate, and severe anxiety, respectively. The Cronbach's alpha of the scale was .92 [24]. It was adapted into Turkish by Konkan et al., and Cronbach's alpha level was calculated as .85 [25]. In our study, Cronbach's alpha value on the scale was .90.

### Procedures performed within the scope of adapting the scale to Turkish

**Linguistic equivalence.** The original questionnaire was translated by two native Turkish translators who can speak fluent English. As a result of the agreement between the translators, the first Turkish version of the PMSS (PMSS-TR) was obtained. Then, two native speakers with fluent Turkish skills were asked to translate the PMSS-TR back into English. The received English back translations were compared with the original scale by the authors, and inconsistencies in the first version of the PMSS-TR were corrected.

The obtained PMSS-TR was applied to 10 students as a pilot. Participants read the questions and verbally evaluated the intelligibility of the items. In line with the feedback received, no changes were required in the scale items. These students were not included in the study.

In terms of linguistic equivalence, the PMSS-English original scale (PMSS-EN) and PMSS-TR were applied crosswise to two groups (Group 1: PMSS-EN followed by PMSS-TR, Group 2: PMSS-TR followed by PMSS-EN) consisting of 35 medical students each (native speakers of Turkish and those studying in the English-instructed medical school program). The relationship between the total scores was examined. In addition, in both versions, free text comments were requested from the participants. In line with the students' feedback, grammatical errors were also reviewed, and the PMSS-TR was given its final form. Afterwards, the PMSS-TR was presented to the expert committee consisting of a public health specialist, two medical education specialists and a psychiatrist. A standard evaluation form was used in the expert panel. In this form, there was an area where opinions were written for each item of the scale and a response section with three options: "appropriate", "can be changed in line with suggestions", and "must be revised". The evaluations made by the experts independently from each other were reviewed, and the PMSS-TR was given its final form.

### Statistical analyses

SPSS 22.0 (IBM, Armonk, NY, USA) and AMOS 24 (IBM) statistical package programs were used for the reliability and validity analyses of the scale. Descriptive statistics are presented as the mean ± standard deviation (SD) for ordinal data and as numbers and percentages for categorical data. Normality tests of ordinal variables and scale scores were performed using Z values calculated for graphing methods, Kolmogorov–Smirnov test, skewness, and kurtosis coefficients. The scores obtained from the PMSS-TR and GAD-7 scales are the mean ± SD. Confirmatory factor analysis (CFA) was used in the validity analysis. In addition, Cronbach's

alpha coefficient was calculated for reliability analysis, and intraclass correlation was checked in the test-retest. A *P* value of < .05 was considered significant.

## Results

### Characteristics of participants

A total of 612 volunteer preclinical medical students participated in the survey. The overall response rate was 56% (population = 1080). A total of 574 students' data were analyzed, excluding 38 questionnaires that were not completed appropriately. The mean age of the participants was 20.9±2.2 years, and 292 were females (50.9%). The sociodemographic characteristics of the study group are presented in Table 1.

### Linguistic equivalence

The original PMSS and PMSS-TR were applied crosswise and separately to two groups of 35 students who were fluent in both languages (trained in the English Medicine program). There was a highly significant correlation between total scores in the first group (PMSS-EN-PMSS-TR) and the second group (PMSS-TR-PMSS-EN) (r = .84, *P* < .001 and r = .81, *P* < .001, respectively).

### Pilot application, ıtem fit ındices, and reliability

A pilot study was conducted with 52 students to determine the internal consistency coefficient, and item correlation fits. The Cronbach's alpha for pilot data was .80. Although items 1, 2, and 11 appeared problematic according to item fit indices, they were not deleted because there was no significant change in Cronbach's alpha value when removed.

**Table 1. Sociodemographic characteristics of the participants.**

|  | Study Year | | |
| --- | --- | --- | --- |
| **Variables** | **Term I** | **Term II** | **Term III** |
| Age [Mean (±SD)] | 21.0 (±2.3) | 20.8 (±2.4) | 21.0 (±1.8) |
| Sex [n (%)] | | | |
| Male | 101 (51.3) | 92 (47.9) | 89 (48.1) |
| Female | 96 (48.7) | 100 (52.1) | 96 (51.9) |
| Income status [n (%)] | | | |
| Good | 33 (16.8) | 41 (21.4) | 38 (20.5) |
| Moderate | 155 (78.7) | 137 (71.4) | 138 (74.6) |
| Bad | 9 (4.6) | 14 (7.3) | 9 (4.9) |
| Tobacco use [n (%)] | | | |
| Yes | 27 (13.7) | 33 (17.2) | 22 (11.9) |
| No | 170 (86.3) | 159 (82.8) | 163 (88.1) |
| Exercise [n (%)] | | | |
| Yes | 106 (53.8) | 108 (56.3) | 97 (52.4) |
| No | 91 (46.2) | 84 (43.8) | 88 (47.6) |

The mean PMSS-TR score of the students was 39.80±8.09, and the GAD-7 score was 11.0±5.5. According to the GAD-7 scale, 28.2% of the students showed mild anxiety, 30.5% moderate, and 28.2% severe anxiety symptoms. There was no significant relationship between the PMSS-TR and GAD-7 scores of the participants and their age, sex, tobacco use, exercise, or income status (*P*>.05).

**Table 2. Item fit indices and cronbach alpha values calculated in the main administration.**

| Item number | Item Total Correlation | Cronbach's Alpha if Item Deleted |
|---|---|---|
| 1 | .50 | .80 |
| 2 | .40 | .81 |
| 3 | .49 | .80 |
| 4 | .43 | .81 |
| 5 | .35 | .81 |
| 6 | .55 | .80 |
| 7 | .39 | .81 |
| 8 | .59 | .80 |
| 9 | .49 | .80 |
| 10 | .57 | .80 |
| 11 | .32 | .82 |
| 12 | .52 | .80 |
| 13 | .31 | .82 |

The reliability analysis of the Turkish form was carried out on the data obtained from 574 students. Item total score correlations were between .31 and .56, and Cronbach's alpha value was .81. A significant difference was found between the upper and lower 27% group scores for all scale items ($P < .001$). The fit indices of items 1, 2, and 11, which seemed problematic in the pilot administration, were also sufficient (Table 2).

A significant correlation was found between each item score and its dimension, as well as between dimensions and the total scale score ($P < .001$ for all). The results of the correlation analysis between the scale items' subdimensions and the total score are given in Table 3.

The number of items, mean score and Cronbach's alpha values for the subdimensions and the overall scale are given in Table 4.

**Table 3. Correlation values between scale items, subdimensions, and the total score.**

| | Psychological Stress and Environment | Resilience and Expectations | Total |
|---|---|---|---|
| Item 1 | .60* | .37* | .59* |
| Item 2 | .39* | .79* | .53* |
| Item 3 | .47* | .65* | .56* |
| Item 4 | .52* | .34* | .51* |
| Item 5 | .30* | .66* | .44* |
| Item 6 | .64* | .44* | .64* |
| Item 7 | .54* | .20* | .49* |
| Item 8 | .66* | .39* | .64* |
| Item 9 | .59* | .30* | .56* |
| Item 10 | .70* | .34* | .65* |
| Item 11 | .42* | .16* | .38* |
| Item 12 | .62* | .31* | .60* |
| Item 13 | .34* | .15* | .46* |
| Psychological Stress and Environment | | .53* | .95* |
| Resilience and Expectations | | | .71* |

* $P<0.001$.

**Table 4. The number of items, mean scores, and Cronbach's alpha values for subdimensions and overall PMSS-TR.**

| PMSS-TR | Number of Items | Mean±SD | Cronbach's Alfa |
|---|---|---|---|
| Psychological Stress and Environment | 9 | 1.95±0.02 | .78 |
| Resilience and Expectations | 3 | 2.46±0.16 | .53 |
| **Overall** | 13 | 2.06±0.05 | .81 |

## Construct validity

Confirmatory factor analysis (CFA) was performed to confirm the original structure of the scale. Meanwhile, item 13 was excluded, as suggested by the developers of the scale, and confirmatory factor analysis was conducted based on the two-factor structure [6]. The fit indices of the model obtained were at an acceptable level after the modifications applied between error variances (CMIN/DF = 1.51. p < .001, RMSEA = .04, CFI = .97, GFI = .97). In confirmatory factor analysis, the two-factor structure of the original scale was confirmed (Fig 1). Acceptable values for model fit indices and values obtained from the model are shown in Table 5.

## Convergent validity

The GAD-7 scale was used for convergent validity. Evidence for convergent validity was sought by evaluating the relationship between students' scores on the PMSS-TR and GAD-7

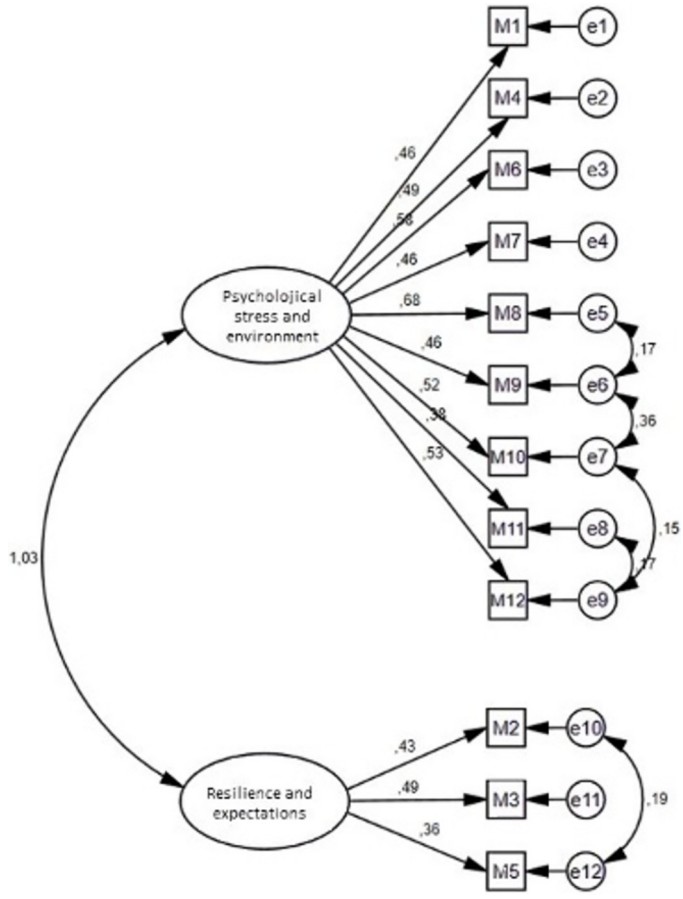

**Fig 1. Confirmatory factor analysis path diagram.**

**Table 5. Acceptable limits of model fit indices and values obtained from the model.**

| Fit Index | Good fit | Acceptable fit | PMSS-TR | Assessment of fit |
|---|---|---|---|---|
| CMIN/DF | $0 \leq$ CMIN/DF $\leq 2$ | $2 \leq$ CMIN/DF $\leq 3$ | 1.51 | Good |
| AGFI | $.90 \leq$ AGFI $\leq 1.00$ | $.85 \leq$ AGFI $\leq .90$ | .95 | Good |
| GFI | $.95 \leq$ GFI $\leq 1.00$ | $.90 \leq$ GFI $\leq .95$ | .97 | Good |
| NFI | $.95 \leq$ NFI $\leq 1.00$ | $.90 \leq$ NFI $\leq .95$ | .91 | Acceptable |
| CFI | $.95 \leq$ CFI $\leq 1.00$ | $.90 \leq$ CFI $\leq .95$ | .97 | Good |
| IFI | $.95 \leq$ IFI $\leq 1.00$ | $.90 \leq$ IFI $\leq .95$ | .97 | Good |
| RMSEA | $.00 \leq$ RMSEA $\leq .05$ | $.05 \leq$ RMSEA $\leq .08$ | .04 | Good |
| SRMR | $.00 \leq$ SRMR $\leq .05$ | $.05 \leq$ SRMR $\leq .10$ | .05 | Acceptable |

$\chi^2(48) = 72.567$, $P = .013$.

AGFI: Adjusted Goodness-of-Fit Index; CFI: Comparative Fit Index; CMIN/df: Chi-Square/Degree of Freedom; GFI: Goodness-of-Fit Index; IFI: Incremental Fit Index; NFI: Normed Fit Index; RMSEA: Root Mean Square Error of Approximation; SRMR: Standardized Root Mean Square Residual.

scales and whether the scores differed significantly according to various sociodemographic characteristics (sex, tobacco use, income status, and exercise). There was a positive and significant relationship between the scales' total scores (r = .48, $P < .001$). However, there was no significant relationship between PMSS-TR and GAD-7 scores and age, sex, income status, tobacco use, or exercise characteristics ($P > .05$).

The scale was administered to a separate group of 70 students 15 days later for test-retest reliability. Intraclass correlation coefficient (ICC) scores ranged from .68 to .88 (Table 6).

## Discussion

Our study examined the Turkish validity and reliability of the PMSS Scale. This scale, developed by Vitaliano et al., is a scale for determining the stress specific to medical school and was defined by the authors as a valid and reliable tool. This study showed that the Turkish

**Table 6. Intraclass correlation coefficients, item and scale score descriptors in test-retest application.**

| Scale dimensions | Items | r (95% CI) | Mean (±SD) | |
|---|---|---|---|---|
| | | Test-Retest Reliability | Test | Retest |
| Psychological Stress and Environment | 1 | .80 (.68–.88) | 2.0 (±1.2) | 1.9 (±1.0) |
| | 4 | .73 (.57–.83) | 2.5 (±1.1) | 2.3 (±1.0) |
| | 6 | .68 (.48–.80) | 2.1 (±1.2) | 2.1 (±1.1) |
| | 7 | .83 (.72–.89) | 2.2 (±1.2) | 2.0 (±1.1) |
| | 8 | .80 (.65–.89) | 2.3 (±1.0) | 1.9 (±1.0) |
| | 9 | .78 (.65–.86) | 2.1 (±1.0) | 2.1 (±1.0) |
| | 10 | .86 (.78–.91) | 1.6 (±1.1) | 1.7 (±1.1) |
| | 11 | .76 (.62–.85) | 1.6 (±1.0) | 1.8 (±1.0) |
| | 12 | .79 (.65–.87) | 1.0 (±1.0) | 1.3 (±1.0) |
| Resilience and Expectations | 2 | .77 (.63–.86) | 2.0 (±1.3) | 2.1 (±1.2) |
| | 3 | .78 (.65–.87) | 2.2 (±1.2) | 2.0 (±1.0) |
| | 5 | .83 (.73–.90) | 3.1 (±1.1) | 2.9 (±1.1) |
| | 13 | .88 (.80–.92) | 1.7 (±1.4) | 1.8 (±1.3) |
| | Total | .92 (.87–.95) | 24.6 (±7.5) | 24.2 (±7.2) |

r: Intraclass correlation coefficient, CI: Confidence interval, SD: Standard deviation.

version of the PMMS scale is an accurate and reliable tool that can be applied to determine perceived medical school-specific stress in Turkish medical students. No original Turkish tool is used to assess medical students' medical school-specific stress. To the best of our knowledge, this is the first study to evaluate medical school-specific stress perceived by medical students in Türkiye.

In the adaptation of the scale, the International Association for Pharmacoeconomics and Outcomes Research guidelines were followed, and the "Principles of Good Practice for the Translation and Cultural Adaptation Process for Patient-Reported Outcomes (PRO) Measure" were applied [28]. In cultural adaptation, it is recommended to pay attention to semantic equivalence and cultural differences instead of the literal translation of scale items [18]. While translating the PMSS scale into Turkish, we considered these suggestions and attached importance to semantic equivalence. For example, since the word "medical school" in the original scale was used as "medical faculty" in Türkiye, its Turkish equivalent was written in this way. A similar approach was used in the German version of the scale [10]. In addition, while scale items were scored between 0–4 in the original scale, most European versions were scored 1–5 [7,10]. We also used the 1–5 scoring system to compare our study results with the European population.

Confirmatory factor analysis is used to determine the validity of measurement tools developed in other samples and cultures [26]. Confirmatory factor analysis was conducted to determine whether the Turkish representative confirmed the scale's factor structure and construct validity. It was observed that each item of the scale contributed significantly to the formation of dimensions in the CFA model. In the current study, a correlation was established between the error variance values of the items in the CFA. For example, items 2 and 5 represent "working conditions and information overload". The correlation of these items was found to be high, and corrections were made. Similarly, items 8 and 9, 9 and 10, and 11 and 12 generally represent the "attitude of the faculties"; the correlation between these items was also found to be high, and corrections were made. As a result, most of the items were combined under the same factors, similar to the original structure. The two-factor structure of the original scale was confirmed in the PMSS-TR, and it was decided to make the naming identical to the original scale. Nine items collected in the first dimension were associated with the "Psychological Stress and Environment" subdomain of the scale. In contrast, three under the second dimension were associated with the "Resilience and Expectations" dimension. Item 13 is finance-related and is not associated with either factor as in the original scale. Therefore, it is not included in one of the dimensions. However, as it was in the original scale, it was not removed from the scale because the students highly approved it.

A slightly modified version of the PMSS was used by Tyssen R et al. In this form of the scale, the original scale's item about internships was changed, and an item about the accommodation was written instead. The internal consistency of the scale was .78 [7]. In the studies conducted in Norway, three dimensions were identified in the scale: 1) the medical school being cold and threatening, 2) concerns about work and competence, and 3) concerns about finance and housing [7,20]. Our study did not change the original scale's items and obtained two dimensions as in the original scale. The difference in the studies mentioned above is the writing of an article about accommodation. In these studies, this item change may have affected the scale's three-factor structure.

According to the results of the goodness-of-fit analysis of the first-level CFA model, the sample model is consistent and significant with the original structure of the scale (CMIN/DF = 1.51, RMSEA = .04, NFI = .91). In our study, the CFI value of the PMSS-TR was .97. The NFI showed an acceptable fit. In contrast, the CMIN-DF, GFI, CFI, and RMSEA showed excellent fits.

Item correlations and Cronbach's alpha were evaluated for the internal consistency of the PMSS-TR. Item total score correlations were between .35 and .59, and Cronbach's alpha value was .81. These results indicate a high level of reliability [27]. High reliability was also found in the adaptation studies of the scale to other languages [7,10].

In our study, Cronbach's alpha values for the scale dimensions were .79 and .63 for the "Psychological stress and environment" and "Resilience and expectations" dimensions, respectively. There are three items in the "Resilience and expectations" dimension of the scale. This may have caused low Cronbach's alpha values. In the current study, we found a significant correlation between each item and its dimension and between the dimensions and the scale's total score. This finding supports that items and dimensions were marked consistently.

Test-retest reliability was significant for all scale items, and the temporal consistency of the scale was good. Since there was no significant change in Cronbach's alpha value when the item was deleted, removing any item from the scale was unnecessary. In our study, item reliability was high for all scale items. There was a significant correlation between the score of each item and the dimension it is in, as well as the dimensions and the scale's total score.

Our study found a significant positive relationship between the total scores of the PMSS-TR and GAD-7 scales. This finding indicated that both scales measure similar concepts. There was no significant relationship between the total scores of both scales and age, sex, tobacco use, exercise, or income status.

The average PMSS-TR score of the students participating in this study was 39.8±8.0, indicating a high perceived stress level. In a survey conducted by Afshar et al. [28] in Germany in which the German version of the PMSS was used, the stress scores of the students were 37.2 ±8.3 (18–65). This study reported that students had high-stress levels [28]. In a study conducted in Poland, the PMSS-PL score was 36.4±8.4 (13–65) [21]. In the study by Tyssen et al. in Norway, students' stress scores were 30.7±7.6 [7]. Similarly, in a study conducted in Germany [10], students' stress levels were higher than in Norwegian studies [7,29]. These results were attributed to the difference in working conditions and the fact that Norwegian doctors were more satisfied with their requirements. Our study's scores were higher than the stress scores obtained in the German and European studies. We think that various factors, such as the education program's structure, the curriculum's density, measurement and evaluation systems, social opportunities, and physicians' working conditions, may have contributed to our students' higher perceived stress levels. Students' financial situation and economic well-being may also have impacted the results. The factors that cause our students to perceive a high-stress level should be clarified with larger, multicenter, and prospective studies.

In the study of Tyssen et al., no difference was found between male and female students regarding general stress levels [7]. Likewise, our research showed no difference between the sexes regarding stress scores.

In this study, the validity and reliability of the Turkish version of the PMSS scale were assessed and found to be acceptable. The PMSS-TR is not a diagnostic tool but an easy-to-apply, practical screening tool that can be used to suspect students' perceived medical school stress. In this respect, it is thought that it will contribute to researchers in Turkish-speaking countries.

## Study limitations

Our study has several limitations. First, it is a methodologic study, and its sample consists of medical students studying at a State University in Türkiye. Therefore, the results cannot be generalized to medical students. Second, the data are based on students' self-evaluations. Although the study population has similar demographic characteristics such as age and

socioeconomic status, the 56% participation rate may have caused a selection bias. The strength of our study is that it provided a valid and reliable tool to investigate medical school-induced stress in medical students in Türkiye. It is the first study in Türkiye to evaluate medical students' stress specific to medical school.

## Conclusion

The PMSS-TR is a valid and reliable tool that can be used alone or in combination with other measurement tools to assess the medical school-related stress of medical students in Türkiye. The scale is a unique, easy-to-answer, practical tool to evaluate stress perceived by medical students in Turkish-speaking communities. It can be used to screen and monitor the stress situations of medical students in Türkiye and to evaluate preventive or mitigating interventions. Medical schools should be aware of the stress of their students caused by medical school. Including stress-reducing strategies in medical school curricula, receiving feedback from students, regular monitoring of students, and early detection of stress and its sources are important for preventive actions. Additionally, future studies should focus on prospective and large-scale research into developing preventive strategies and the effectiveness of interventions using the PMSS-TR.

## Supporting information

**S1 Dataset.**
(SAV)

**S2 Dataset.**
(SAV)

## Acknowledgments

We thank Professor Peter P. Vitaliano for his cooperation in validating the Perceived Medical School Stress Scale in Turkish medical students. We also thank the medical students who participated in the study.

## Author Contributions

**Conceptualization:** Esra Çınar Tanrıverdi, Sinan Yılmaz, Yasemin Çayır.

**Data curation:** Esra Çınar Tanrıverdi, Yasemin Çayır.

**Formal analysis:** Esra Çınar Tanrıverdi, Sinan Yılmaz.

**Methodology:** Esra Çınar Tanrıverdi, Sinan Yılmaz, Yasemin Çayır.

**Resources:** Esra Çınar Tanrıverdi, Yasemin Çayır.

**Supervision:** Esra Çınar Tanrıverdi, Sinan Yılmaz, Yasemin Çayır.

**Validation:** Esra Çınar Tanrıverdi, Yasemin Çayır.

**Visualization:** Esra Çınar Tanrıverdi.

**Writing – original draft:** Esra Çınar Tanrıverdi, Yasemin Çayır.

**Writing – review & editing:** Esra Çınar Tanrıverdi, Sinan Yılmaz, Yasemin Çayır.

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
