## [Decision Letter · Decision Letter 0]

14 Feb 2023

PONE-D-22-28252Measurement of medical faculty-specific stress: Validity and Reliability of the Turkish Version of the Perceived Medical School Stress (PMSS) Scale PLOS ONE

Dear Dr. Çınar Tanrıverdi,

Thank you for submitting your manuscript to PLOS ONE. After careful consideration, we feel that it has merit but does not fully meet PLOS ONE’s publication criteria as it currently stands. Therefore, we invite you to submit a revised version of the manuscript that addresses the points raised during the review process.

We look forward to receiving your revised manuscript.

Kind regards,

Somayeh Delavari, Ph.D.,

Academic Editor

PLOS ONE

Journal Requirements:

2**. **Please include captions for your Supporting Information files at the end of your manuscript, and update any in-text citations to match accordingly. Please see our Supporting Information guidelines for more information: http://journals.plos.org/plosone/s/supporting-information. 

Additional Editor Comments (if provided):

Reviewers' comments:

Reviewer's Responses to Questions

**Comments to the Author**

1. Is the manuscript technically sound, and do the data support the conclusions?

Reviewer #1: Yes

Reviewer #2: Yes

Reviewer #3: Yes

Reviewer #4: Yes

2. Has the statistical analysis been performed appropriately and rigorously? 

Reviewer #1: Yes

Reviewer #2: Yes

Reviewer #3: Yes

Reviewer #4: Yes

3. Have the authors made all data underlying the findings in their manuscript fully available?

Reviewer #1: Yes

Reviewer #2: Yes

Reviewer #3: Yes

Reviewer #4: Yes

4. Is the manuscript presented in an intelligible fashion and written in standard English?

Reviewer #1: Yes

Reviewer #2: Yes

Reviewer #3: Yes

Reviewer #4: Yes

5. Review Comments to the Author

Reviewer #1: Dear Aouthers:

thank you for conducting this valuable study. This study is about the one of important problems in teaching and learning of medical students. However, after reviewing the manuscript, I believe there are important issues which need to be addressed before the manuscript is ready for publication in PLOS ONE. My comments (both editorial and substantive) are found below:

1- The main work in this study is evaluating validity and reliability of PMSS that I think the first sentence in the topic “Measurement of medical faculty-specific stress” can be deleted.

2- In the topic, Please mention that in whose population you want evaluate validity and reliability of PMSS.

3- One of the stages of evaluating validity and reliability of scales is Qualitative Content Validity. (Face validity for checking difficulty, relevance and ambiguity, Content validity for checking grammar, wording, Item location and scaling). I do not see this stage in your research.

4- Test-retest reliability was performed in the pilot stage?

5- What is the means of “Test-retest reliability was significant for all items”? In the Test-retest reliability we expect that the significant change wasn’t happened after 15 days period and the most important thigs is the correlation coefficient of scales in two period.

6- I suggest that use ICC (Intra Class Correlation) to evaluation of correlation coefficient In the Test-retest reliability.

7- Please report the Chi-Squared P-Value indices.

8- I think your means of "Criterion validity" in the assessment of Construct validity are "Convergent Validity". Because in the Convergent Validity we use another scales that validated before to confirm construct validity of new scale.

9- Why the researchers didn’t use “Perceived Stress Scale” for Convergent Validity?

Good Luck

Reviewer #2: The authors did not add item 13 to one of the scale dimensions as in the original article of the scale. In the method section, they should explain this situation with reference to the original article.

Can ICC (intraclass correlation coeffience) result be presented in test-retest comparison for scale sub-dimensions and total score?

Cronbach's alpha values for the sub-dimensions should be presented in a table in the findings section (presented in the discussion section). Having 3 items in a dimension may cause a low alpha value. This result can be explained in this way in the discussion section of the article.

The presentation of the results of the pilot study in Table 2 causes confusion. The results of the pilot study should be omitted from table 2 and mentioned in the text only.

In the CFA analysis, a connection was established between the error variance values of the items. If such a correction is to be made in the model, the reason for this should be briefly explained in the discussion section. for example, "items represent 'too much course load' for the connection between 2 and 5, so the correlation between them was found to be high and correction was made". Otherwise, it appears that extra effort is spent in statistical analysis for the model's good fit value.

Reviewer #3: The general research question is a valid and the authors answers to it appropriately. However, there are some comments for improving the manuscript.

- The participation rate in this study is reported to be 65%, which can be a type of selection bias. Were those who participated in the study different from those who did not participate in terms of important variables such as (demographics, socioeconomic status, etc.) to assess the validity and reliability of a questionnaire, the selection of participants should be random for generalizability.

Reviewer #4: Thank you for giving me this opportunity to review the paper. The study design and sample size are acceptable and the paper is well-written by and large. I have a few comments:

1. Maximum number of both scales should be added to the abstract.

2. 'Medical Faculty' is not an appropriate key-word.

3. What is 'XXX' in line 94?

4. A " is missed in line 135.

5. Psychometric properties of the original version should be added.

6. The manuscript needs exact proof reading for any typo like '(24]'.

7. What is definition of 'Income status'?

6. PLOS authors have the option to publish the peer review history of their article (what does this mean?). If published, this will include your full peer review and any attached files.

Reviewer #1: No

Reviewer #2: No

Reviewer #3: No

Reviewer #4: **Yes: **Mohammadreza Shalbafan

---

## [Author Response · Author response to Decision Letter 0]

23 Feb 2023

Dear Reviewer,

We would like to thank you for your insightful comments and suggestions. We made all possible changes that were suggested and detailed the changes in the table below. Prior to responding to your comments, we want to inform you that all the revisions and improvements are highlighted red in the revised version of our manuscript. We sincerely appreciate your comments on our manuscript. We would like to thank you again for your valuable time and insight to strengthen our paper.

Yours truly,

Corresponding author on behalf of the authors.

 Reviewer #1 Comment and Response Reviewer #1

1. The main work in this study is evaluating validity and reliability of PMSS that I think the first sentence in the topic “Measurement of medical faculty-specific stress” can be deleted.

We have removed “Measurement of medical faculty-specific stress” from the title.

2. In the topic, Please mention that in whose population you want evaluate validity and reliability of PMSS.

The title has been amended as follows: “Assessing the validity and reliability of the Perceived Medical School Stress Scale in Turkish medical students”.

3. One of the stages of evaluating validity and reliability of scales is Qualitative Content Validity. (Face validity for checking difficulty, relevance and ambiguity, Content validity for checking grammar, wording, Item location and scaling). I do not see this stage in your research.

Necessary explanations have been added to the Linguistic equivalence section.

(Afterwards, PMSS-TR was presented to the expert committee consisting of a public health specialist, two medical education specialists and a psychiatrist. A standard evaluation form was used in the expert panel. In this form, there was an area where opinions were written for each item of the scale and a response section with three options as “appropriate”, “can be changed in line with suggestions” and “must be revised”. The evaluations made by the experts independently from each other were reviewed and PMSS-TR was given its final form.)

4. Test-retest reliability was performed in the pilot stage?

Test-retest reliability was not performed in the pilot stage.

5. What is the means of “Test-retest reliability was significant for all items”? In the Test-retest reliability we expect that the significant change wasn’t happened after 15 days period and the most important thigs is the correlation coefficient of scales in two period.

An expression error has occurred in this field. What is meant to be expressed is that the relationships between consecutive applications are positive and significant for all items.

We have made the relevant corrections. This finding is presented in Table 2 by ICC analysis.

6. I suggest that use ICC (Intra Class Correlation) to evaluation of correlation coefficient In the Test-retest reliability.

ICC results were added to the study (Table 2).

7. Please report the Chi-Squared P Value indices.

We have added Chi-Squared and P Value. (Table 6)

8. I think your means of "Criterion validity" in the assessment of Construct validity are "Convergent Validity". Because in the Convergent Validity we use another scales that validated before to confirm construct validity of new scale.

Corrected as "Convergent Validity" throughout the article.

9. 

Why the researchers didn’t use “Perceived Stress Scale” for Convergent Validity?

Although the “Perceived Stress Scale” measures perceived stress, it does not show stress specific to medical school.

Anxiety is identified as the main type of distress among medical students.

Developing the original scale, Vitalino et al., it has been reported in previous studies of medical school-induced stress that there is a significant correlation with anxiety. (Vitaliano PP, Russo J, Carr JE, Heerwagen JH. Medical school pressures and their relationship to anxiety. Journal of Nervous and Mental Disease. 1984- reference 6)

For convergent validity, we preferred to use the GAD-7, a widely used anxiety scale with proven validity and reliability in Turkey, in addition to the PMSS scale, since there is no instrument that measures stress specific to medical school and is adapted to Turkish.

GAD-7 scores showed significant correlation with PMSS scores.

However, we do not think that GAD-7 is the gold standard in ensuring convergent validity.

Reviewer #2 Comment and Response Reviewer #2

1. The authors did not add item 13 to one of the scale dimensions as in the original article of the scale. In the method section, they should explain this situation with reference to the original article.

In the method section, 132-134. specified in the line.

(The 13th item was added to the scale by the authors' majority approval because it was a primary concern of students (finance). However, this item was not included in either of the two subdimensions)

2. Can ICC (intraclass correlation coeffience) result be presented in test-retest comparison for scale sub-dimensions and total score?

The ICC (intraclass correlation coefficient) result in the test-retest comparison for the scale sub-dimensions and the total score is shown in Table 2.

3. Cronbach's alpha values for the sub-dimensions should be presented in a table in the findings section (presented in the discussion section).

Cronbach's alpha values for the sub-dimensions were presented in a Table in the findings section (Table 5 has been added).

4. Having 3 items in a dimension may cause a low alpha value. This result can be explained in this way in the discussion section of the article.

This result explained in this way in the discussion section of the article.

5. The presentation of the results of the pilot study in Table 2 causes confusion. The results of the pilot study should be omitted from table 2 and mentioned in the text only.

The results of the pilot study were deleted from Table 2 (named Table 3 after revision) and cited in the text only.

6. In the CFA analysis, a connection was established between the error variance values of the items. 

If such a correction is to be made in the model, the reason for this should be briefly explained in the discussion section. for example, "items represent 'too much course load' for the connection between 2 and 5, so the correlation between them was found to be high and correction was made". 

Otherwise, it appears that extra effort is spent in statistical analysis for the model's good fit value.

In the current study, a correlation was established between the error variance values of the items in the CFA analysis.

For example, 2 and 5 represent “working conditions and information overload”. The correlation of these items was found to be high and correction was made. Similarly, items 8 and 9, items 9 and 10, items 11 and 12 generally represent the "attitude of the faculties", the correlation between these items was also found to be high and correction was made.

Added to the discussion.

Reviewer #3 Comment and Response Reviewer #3

1.The participation rate in this study is reported to be 65%, which can be a type of selection bias.

Were those who participated in the study different from those who did not participate in terms of important variables such as (demographics, socioeconomic status, etc.) to assess the validity and reliability of a questionnaire, the selection of participants should be random for generalizability.

The study population has similar characteristics in terms of demographic characteristics such as age and socioeconomic status.

However, the 65% participation rate in the research may have caused a selection bias. The fact that the participants were not randomly selected in our study is one of the limitations of the study.

We have added this statement to the limitations.

Reviewer #4 Comment and Response Reviewer #4

1.Maximum number of both scales should be added to the abstract.

We have added more information about PMSS and GAD-7 to the abstract section.

2.'Medical Faculty' is not an appropriate key-word.

We have removed 'Medical Faculty' from keywords.

3.What is 'XXX' in line 94?

XXX was written for blinding. All changed to “Atatürk”.

4. A " is missed in line 135.

Corrected.

5. Psychometric properties of the original version should be added.

We have added psychometric properties and internal consistency of the original version.

6. The manuscript needs exact proof reading for any typo like '(24]'

The entire manuscript has been revised for mistypos, if any

7. What is definition of 'Income status'?

Students were asked about their own perceptions of their economic situation. They were asked to choose one of the definitions as good, medium or bad.

---

## [Decision Letter · Decision Letter 1]

30 Mar 2023

PONE-D-22-28252R1Assessing the validity and reliability of the Perceived Medical School Stress Scale in Turkish medical studentsPLOS ONE

Dear Dr. Çınar Tanrıverdi,

Thank you for submitting your manuscript to PLOS ONE. After careful consideration, we feel that it has merit but does not fully meet PLOS ONE’s publication criteria as it currently stands. Therefore, we invite you to submit a revised version of the manuscript that addresses the points raised during the review process.

We look forward to receiving your revised manuscript.

Kind regards,

Somayeh Delavari, Ph.D.,

Academic Editor

PLOS ONE

Journal Requirements:

Reviewers' comments:

Reviewer's Responses to Questions

**Comments to the Author**

1. If the authors have adequately addressed your comments raised in a previous round of review and you feel that this manuscript is now acceptable for publication, you may indicate that here to bypass the “Comments to the Author” section, enter your conflict of interest statement in the “Confidential to Editor” section, and submit your "Accept" recommendation.

Reviewer #1: All comments have been addressed

Reviewer #2: All comments have been addressed

Reviewer #3: (No Response)

Reviewer #4: (No Response)

2. Is the manuscript technically sound, and do the data support the conclusions?

Reviewer #1: Yes

Reviewer #2: Yes

Reviewer #3: Yes

Reviewer #4: (No Response)

3. Has the statistical analysis been performed appropriately and rigorously? 

Reviewer #1: Yes

Reviewer #2: Yes

Reviewer #3: No

Reviewer #4: (No Response)

4. Have the authors made all data underlying the findings in their manuscript fully available?

Reviewer #1: Yes

Reviewer #2: Yes

Reviewer #3: Yes

Reviewer #4: (No Response)

5. Is the manuscript presented in an intelligible fashion and written in standard English?

Reviewer #1: Yes

Reviewer #2: Yes

Reviewer #3: Yes

Reviewer #4: (No Response)

6. Review Comments to the Author

Reviewer #1: Hello dear authors

Thank you for considering the comments in revised article.

Just for imprving manuscript I suggest that using "Evauation" instesd of "Assessing" in topic.

Table 2 are very confusing. unfortunatelly I couldnt understand what information have presented in this table (Test-re-test reliabilty or ICC).

If you didnt calculate Test-re-test reliabilty andICC for pilot Application, I suggest that seperate this section from pilot aplication section and present that after convergent validity. Absolutely we should analyse the reliability of quastionnaire after assuring of vlidity.

thank you

Reviewer #2: (No Response)

Reviewer #3: Dear authors

The most important point to evaluate the validity and reliability of the questionnaire is generalizability. Although the authors stated that the demographics and underlying information of the participants were similar, they had to show statistically that there was no difference between the respondents' responses and those who did not respond. This can reduce the selection bias to some extent.

My concern about the selection bias has not been addressed.

Reviewer #4: Thank you for revisions. My comments have been addressed appropriately. No further comment from my side.

7. PLOS authors have the option to publish the peer review history of their article (what does this mean?). If published, this will include your full peer review and any attached files.

Reviewer #1: No

Reviewer #2: No

Reviewer #3: No

Reviewer #4: **Yes: **Mohammadreza Shalbafan

---

## [Author Response · Author response to Decision Letter 1]

15 Jun 2023

Dear Reviewers,

We would like to thank you for your insightful comments and suggestions. We made all possible changes that were suggested and detailed the changes in the table below. Prior to response your comments, we want to inform you that all the revisions and improvements are highlighted red in the revised version of our manuscript. We sincerely appreciate your comments on our manuscript. We would like to thank you again for your valuable time and insight to strengthen our paper.

Yours truly,

Corresponding author on behalf of the authors.

Reviewer 1 Comment

1. I Thank you for considering the comments in revised article. Just for imprving manuscript I suggest that, using "Evauation" instesd of "Assessing" in topic.

We changed “assessing” as “evalaution” in the title as you recommend. 

2. Table 2 are very confusing. unfortunatelly I couldnt understand what information have presented in this table (Test-re-test reliabilty or ICC).

Table 2 shows test-retest reliability.

 We didn’t calculate test-retest reliability and ICC for pilot application. 

 For this reason, we seperated the ICC from pilot application section and it has been added to after convergent validity as you suggest.

Reviewer 3 Comment

1. The most important point to evaluate the validity and reliability of the questionnaire is generalizability. Although the authors stated that the demographics and underlying information of the participants were similar, they had to show statistically that there was no difference between the respondents' responses and those who did not respond. This can reduce the selection bias to some extent.

My concern about the selection bias has not been addressed.

Thank you for your feedback regarding my article and for highlighting your concern regarding the potential selection bias in evaluating the validity and reliability of the questionnaire. I apologize if my previous response did not adequately address your concern. 

While I did mention that the demographics and underlying information of the participants were similar, I agree that it is important to statistically demonstrate that there were no significant differences between the respondents' responses and those who did not respond. Unfortunately, due to certain limitations, such as data availability, I was unable to perform a statistical comparison between respondents and non-respondents in this particular study. Furthermore we addressed this limitation in the article. Your critical input is invaluable in helping me recognize the gaps in my research and improve its quality. Thank you once again for your valuable feedback and for your interest in my research.

---

## [Decision Letter · Decision Letter 2]

4 Jul 2023

Evaluation the validity and reliability of the Perceived Medical School Stress Scale in Turkish medical students

PONE-D-22-28252R2

Dear Dr. Çınar Tanrıverdi,

We’re pleased to inform you that your manuscript has been judged scientifically suitable for publication and will be formally accepted for publication once it meets all outstanding technical requirements.

Kind regards,

Somayeh Delavari, Ph.D.,

Academic Editor

PLOS ONE

---

## [Editor Report · Acceptance letter]

4 Aug 2023

PONE-D-22-28252R2 

Evaluation the validity and reliability of the Perceived Medical School Stress Scale in Turkish medical students 

Dear Dr. Çınar Tanrıverdi:

I'm pleased to inform you that your manuscript has been deemed suitable for publication in PLOS ONE. Congratulations! Your manuscript is now with our production department. 

Kind regards, 

on behalf of

Dr. Somayeh Delavari 

Academic Editor

PLOS ONE